# Surface Properties of Synaptosomes in the Presence of L-Glutamic and Kainic Acids: In Vitro Alteration of the ATPase and Acetylcholinesterase Activities

**DOI:** 10.3390/membranes11120987

**Published:** 2021-12-17

**Authors:** Virjinia Doltchinkova, Nevena Mouleshkova, Victoria Vitkova

**Affiliations:** 1Department of Biophysics and Radiobiology, Faculty of Biology, Sofia University “St. Kliment Ohridski”, 8 Dragan Tsankov Blvd., 1164 Sofia, Bulgaria; veni40@mail.bg; 2Georgi Nadjakov Institute of Solid State Physics, Bulgarian Academy of Sciences, 72 Tsarigradsko Chaussee Blvd., 1784 Sofia, Bulgaria; victoria@issp.bas.bg

**Keywords:** synaptosomes, L-glutamate, kainic acid, electrophoretic mobility, surface electrical charge, enzyme activity, bending elasticity

## Abstract

Morphologically and functionally identical to brain synapses, the nerve ending particles synaptosomes are biochemically derived membrane structures responsible for the transmission of neural information. Their surface and mechanical properties, measured in vitro, provide useful information about the functional activity of synapses in the brain in vivo. Glutamate and kainic acid are of particular interest because of their role in brain pathology (including causing seizure, migraine, ischemic stroke, aneurysmal subarachnoid hemorrhage, intracerebral hematoma, traumatic brain injury and stroke). The effects of the excitatory neurotransmitter L-glutamic acid and its agonist kainic acid are tested on Na^+^, K^+^-ATPase and Mg^2+^-ATPase activities in synaptic membranes prepared from the cerebral cortex of rat brain tissue. The surface parameters of synaptosome preparations from the cerebral cortex in the presence of L-glutamic and kainic acids are studied by microelectrophoresis for the first time. The studied neurotransmitters promote a significant increase in the electrophoretic mobility and surface electrical charge of synaptosomes at 1–4 h after isolation. The measured decrease in the bending modulus of model bimolecular membranes composed of monounsaturated lipid 1-palmitoyl-2-oleoyl-sn-glycero-3-phosphocholine provides evidence for softer membranes in the presence of L-glutamate. Kainic acid does not affect membrane mechanical stability even at ten-fold higher concentrations. Both the L-glutamic and kainic acids reduce acetylcholinesterase activity and deviation from the normal functions of neurotransmission in synapses is presumed. The presented results regarding the modulation of the enzyme activity of synaptic membranes and surface properties of synaptosomes are expected by biochemical and biophysical studies to contribute to the elucidation of the molecular mechanisms of neurotransmitters/agonists’ action on membranes.

## 1. Introduction

The surface electrical charge and mechanical properties of synaptosomes predetermine their functional activity, altering the synapse physiological activity in the presence of neurotransmitters such as L-glutamic (L-Glu) or its agonists, kainic acids (KA). The synapse provides information communication between neurons, as well as between them and effector organs. It is the platform hosting the application sites of the action of drugs, and endogenous and exogenous factors [1,2].

Isolated synaptosomes perform a number of anabolic and catabolic reactions on neurons [3,4]. The key function of nerve terminals is to process information, an activity that is accomplished by moving ions continuously across the plasma membrane [3]. The significance of elucidation of the role of exogenously added glutamate and kainic acid uptake in the surrounding medium of synaptic and of synaptosomal membranes presents a new biophysical event, underlying numerous pathologies, including pathological conditions such as seizure, migraine, traumatic brain injury and stroke [5,6] The enzymatic activities, and electrokinetic and mechanical properties of synaptosomes (“nerve-ending particles”, “nerve terminals”) are intimately related to the central nervous system (CNS) function.

Synaptosomes are a target for amino acids interaction in neurobiological studies [7]. Synaptosomes, as an experimental system, have widespread application in neurochemistry, neuropharmacology and neurotoxicology [7,8,9]. The rapid development of chemistry and pharmacology has led to the synthesis and testing of a huge number of chemical compounds, with direct or indirect influence on the CNS [10]. Despite the success of the study of the effects of such compounds at the organism or organ level, the mechanisms of their action at the subcellular level are not sufficiently elucidated [11].

Synaptosomes, as biochemically derived membrane structures that are morphologically and functionally identical to brain synapses, carry the main property of thinking matter (the brain)—the transmission of neural information [9]. The mechanism of synaptic transmission consists in the mutual conversion of electrical into physical impulses and vice versa. Therefore, the main task of the present work is to develop a complex approach for studying the bioelectrical characteristics of synaptosomes and the connection with their structure and functions. The electrokinetic behavior of synaptosomes as a model system will provide important information about the intimate mechanism of the functioning of the synapses, in order to clarify the synaptic plasticity as a result of changes in the surface membrane properties of synaptosomes and of key membrane enzymes in synaptic membranes.

The physicochemical properties of biological membranes largely determine the functional activity of nerve cells during signal propagation [12]. The interfacial electrostatics are monitored as a highly important parameter in surface membrane properties. L-Glu and KA treatments of synaptosomes during in vitro aging alter their electrokinetic properties (electrophoretic mobility, zeta potential, surface charge density), which provides information about the biochemical microenvironment of vesicular surfaces. It has been established that the environmental condition and elastic properties of biological membranes largely determine the functional activity of nerve cells [13]. The membrane modulus of the other model system—giant unilamellar vesicles (GUVs)—might be of assistance in explaning the membrane’s elastic properties, such as membrane rigidity.

The realization of the specific function of the nervous system is inconceivable without the participation of some key enzymes, such as Na^+^, K^+^-ATPase, the enzymes responsible for the synthesis of neurotransmitters, as well as the enzymes regulating their effect on synapses. Na^+^, K^+^-ATPase act as a proton pump [14], which maintains resting cell membrane potential by pumping sodium and potassium against the electrochemical gradient across a cell membrane for ions [15]. Neurotransmitters affect the activity of Na^+^ and K^+^-ATPase. Hence, these enzymes are important in the transformation of the extracellular signal into the cellular response, being involved in signaling pathways [16].

The membrane enzyme acetylcholinesterase (AChE, EC 3.1.1.7), present in the cholinergic and noncholinergic tissues, is responsible for the hydrolysis of the neurotransmitter in the postsynaptic clefts [17]. The regulation of acetylcholinesterase activity depends on the influence of a number of factors that are located in the area of the synapse and that can change its activity, such as ions, biological active substances, metabolites or physical environmental factors [18]. Given that acetylcholine may affect protein synthesis, energy metabolism and other processes, these effects might have a wide physiological significance.

The cholinergic system has a proven involvement in basic physiological processes such as learning and memory, thermoregulation, emotional behavior, motor activity, etc.; the study of the influence of endogenous substances remains relevant. Glutamate is the most common excitatory neurotransmitter in the CNS and glutamate receptors are found throughout the brain and spinal cord in neurons and glia [19,20]. Glutamate is localized in the cytoplasm without selective accumulation in the synaptic areas. The propensity for a negatively charged glutamate side chain to receive a proton and achieve a neutral state in bilayer membranes has been studied [21]. Glutamate can be found in the interior of certain membrane proteins and, when charged, can alter protein folding [22] and strongly influences the helix tilt [21]. The ionization of the glutamic chain predicts a dependance on its depth of burial in a lipid membrane [21]. Analytical tools for quantitative measurements of glutamate show that an isolated single synaptic vesicle encapsulates about 8000 glutamate molecules and is comparable to the measured exocytotic quantal glutamate release in amperometric glutamate sensing in the nucleus accumbens of mouse brain tissue [23].

The agonist kainic acid (KA) represents a decarboxylated analogue of glutamate in which the pyrrolidine ring restricts the number of conformationals available to the part of the KA molecule analogous to L-glutamic acid. KA has been identified as a weak inhibitor of the “high affinity” uptake of L-Glu into rat brain synaptosomes [24]. The agonist has been found to bind to rat brain membranes in the absence of sodium ions in a manner indicating binding to a population of receptor sites for L-glutamic acid [25]. KA induced excitotoxicity in rodent models is reported to result in seizures, behavioral changes, oxidative stress, and neurodegeneration of the brain upon KA administration [26]. Lipid peroxidation induced by KA in rat brain tissues has been identified [27].

Despite the abundant literature data on the role of some amino acids as mediators, their mechanisms of action still remain not entirely disclosed at the level of the nerve cell and, in particular, the synapse. The pronounced excitatory and cytotoxic effects of glutamate are associated with changes in the properties of synaptic membranes that have not yet been fully elucidated. This may presumably extend to other mediator amino acids.

Taking into account the above considerations, in the present study we probe the effect of L-glutamic acid (glutamate) and its agonist kainic acid on the activity of the enzymes of synaptic membranes, brain ATPases and AChE, in order to acquire knowledge of their functional biochemical in vitro effect relevant to their physiological action.

Neurons that operate on glutamate, gamma aminobutiric acid and acetylcholine are located in the same parts of the brain, namely, the cortex, striatum, cerebellum. The effect of amino acids on such an important enzyme in the cholinergic system could affect the amount of acetylcholine and, thus, have an effect on cholinergic function.

Synaptosomes carry a negative charge at physiological pH [11,28,29,30,31]. Their isoelectric point is determined at pH 4.0 [29]. Mammalian synaptic vesicles contain high levels of cholesterol, phosphatidylcholine, phosphatidylethanolamine, phosphatidylserine and sphingomyelin, a low amount of total phosphatidylinositol and an unusually high degree of highly unsaturated fatty acid chains as poly-unsaturated fatty acids (PUFAs) [32]. PUFAs stabilize the membrane of synaptic vesicles, which is highly curved because of their small diameter, minimizing the exposure of hydrophobic patches [33]. Synaptosomes are amenable to both structural and functional studies of the synapse, because they not only provide sufficient material for protein biochemical experiments, but also can maintain metabolic activity and membrane potential and can be stimulated to release neurotransmitter [9]. In this work, for the first time, we monitor the influence of L-Glu and KA on the electrophoretic mobility of synaptosomes and the surface electrical charge during their incubation in low ionic strength (50 mM Tris-HCl, pH 7.2) suspending media. The ability of synaptosomal membrane to bend is related to the mechanisms of interaction between amino acids (L-Glu, KA) and the lipid bilayer, providing a versatile platform for membrane mediated cellular processes. Many experimental and computational studies have shown that the biological membrane can bend to expose charged and polar residues to the lipid head groups and water, greatly reducing the cost of protein insertion [13,34]. Protein stability depends on membrane stiffness. The membrane charge increases the bending rigidity relative to the charge-free membrane [33,35]. The surface charge is expected to stiffen the membrane due to the repulsion between charged lipids effectively resisting the membrane bending. The variations of the bending rigidity and surface charge have been studied using the giant unilamellar vesicles (GUVs) as a suitable system for systematic measurements on bio-inspired membranes [36]. GUVs with diameters at the micrometer scale are exploited successfully in the biophysical research for modelling the basic physical properties of biomembranes [37,38]. Besides having characteristic diameters in the same range as the typical cell sizes (5–100 µm), GUVs allow controlling numerous parameters, such as membrane composition, viscosity and the concentrations of solutes in the aqueous environment, and, thus, give the opportunity for the direct visualization of particular membrane-related phenomena at the level of single vesicles [39]. In order to study the electrostatic interaction of L-Glu and KA with the synaptosomal membranes, the electrokinetic properties of synaptosomes are measured during their incubation in media containing the studied biologically active substances. The relationship between the surface electrical charge of the synaptosomal membrane and its stability in microelectrophoresis during in vitro aging, on the one hand, and the membrane rigidity, on the other, is determined. By means of thermal shape fluctuation analysis performed on GUVs, we probe for the nonspecific effects of the studied neurotransmitters on the bending elasticity of model phosphatidylcholine membranes for the first time.

The present work explores the effect of L-glutamic and kainic acids on brain ATPases and new data on the acetylcholinesterase of synaptic membranes from cerebral cortex in view of revealing the structural–functional changes in the synapses. The investigation of the enzyme activity, the properties of synaptic membranes, as well as the vesicular regulatory effects of endogenous and exogenous factors, represents an important step toward enhancing the fundamental knowledge on neurotransmitters interaction with membranes, thus allowing for practical intervention in hitherto unresolved problems in biomedicine.

## 2. Materials and Methods

### 2.1. Materials

All chemicals used in the present study are of analytical grade. Sucrose (saccharose); Trizma^®^hydrochloride–Tris [hydroxymethyl] aminomethan hydrochloride, L-glutamic acid, Kainic acid monohydrate (2-Carboxy-3-carboxymethyl-4-isopropenyl-pyrrolidine.H_2_O), Na_2_HPO_4_.2H_2_O (sodium phosphate dibasic dihydrate), NaH_2_PO_4_.H_2_O (sodium phosphate monobasic monohydrate), acetyl-thiocholine iodide, DTNB (5,5′-Dithiobis (2-nitrobenzoic acid), Ezerin Salicylate, TCA (Trichloroacetic acid), EDTA (Ethylenediaminetetraacetic acid, EDTA disodium salt), Ouabain, NaCl, KCl, MgCl_2_ are purchased from Sigma-Aldrich (St. Louis, MO, USA). ATP (adenosine-5′-triphosphate, Na_2_ salt) is purchased from Serva Electrophoresis GmbH (Heidelberg, Germany). The synthetic monounsaturated phospholipid 1-palmitoyl-2-oleoyl-sn-glycero-3-phosphocholine (POPC, Avanti Polar Lipids Inc., Alabaster, AL 35007, USA) is used. Bidistilled water from quartz distiller is used for the preparation of aqueous solution.

### 2.2. Isolation of Synaptosomes

The experiments of the present work are performed with outbred male albino rats weighing of 150 g (about 60 days old). The rats (57 animals) are supplied by the breeding base of the Faculty of Biology–Sofia housed under standard conditions (temperature 22 ± 2 °C, 12 h light/dark schedule, standard food and water ad libitum. The present study complies with the ethical regulations and legislation in both Europe and Bulgaria. The experiments are performed in compliance with “Directive 2010/63/EU of the European Parliament and of the Council of 22 September 2010 on the protection of animals used for scientific purposes [40]. Ether-anesthetized male albino rats (five animals for each experiment) are decapitated and the brains are rapidly removed and dissected on ice-cold Petri dishes. Crude synaptosomal fraction is prepared by cerebrum. The cerebral cortex (of the two cerebral hemispheres) are separated and washed in physiological saline solution at a temperature of 0 °C. Ten percent tissue homogenate (w/v) is prepared in a glass Potter–Elvehjem homogenizer with a Teflon pestle in ice-cold homogenizing medium (0.32 M Sucrose solution containing 0.05 M Tris-HCl, pH 7.4, and 1 mM EDTA). The crude synaptosomal fraction is prepared in the following manner, [8], with modifications. The brain tissue homogenate is centrifuged at 1000× *g* for 10 min on Janetzki K24 (Engelsdorf (Bez. Leipzig), Germany) centrifuge, 6 × 35 mL Winkel rotor. The pellet consisting of nuclei and cell debris is discarded and the supernatant was centrifuged at 17,500× *g* for 20 min. The pellet is re-suspended in 0.32 M Sucrose and re-centrifuged at 17,500× *g* for 20 min. The pellet defined as crude synaptosomal fraction is re-suspended in the incubation buffer. The resulting pellet is re-suspended in 0.32 M sucrose and layered onto a discontinuous sucrose gradient composed of 10 mL 1.2 M sucrose under 10 mL 0.8 M sucrose. The gradient is centrifuged at 55,000× *g* for 2 h on a Beckman flying rotor centrifuge. The light brown band at the 0.8/1.2 M sucrose interfaces is collected as synaptosomes.

The purified synaptosomal fraction of interest, located between the 0.8 M and 1.2 M sucrose layers, is centrifuged at 17,500× *g* for 30 min on a Janetzki K24, Engelsdorf (Bez. Leipzig), Germany. The resulting pellets are suspended in 0.32 M sucrose and stored at −20 °C if not used immediately.

### 2.3. Determination of Acetylcholinesterase Activity

The spontaneous reaction between the sulfhydryl radical of thiocholine and DTNB (di, thio-bis-nitrobenzoic acid) is used to give a yellow colored 5-mercapto-p-nitrobenzoate. A 0.1 M sodium phosphate buffer (Na_2_HPO_4_·2H_2_O/NaH_2_PO_4_·H_2_O, pH 8.0; 0.075 M acetyl-thiocholine iodide; DTNB 0.01 M; 0.01 M ezerin salicylate) is used [41]. At the concentrations used, the reagents do not inhibit cholinesterase. The samples are measured spectrophotometrically at λ = 412 nm. The calculations used the molar extinction coefficient of 5-mercapto-p-nitrobenzoate (1.35 × 10^4^), measured under standard reaction conditions.

### 2.4. Determination of the Enzymatic Activity of Brain ATPases

The enzyme activity of ATPase is determined by the increase in the amount of inorganic phosphate in the samples during their incubation [42]. The composition of the incubation medium for determining the activity of Na^+^, K^+^, Mg^2+^-ATPase (total ATPase) contains: 0.05 M Tris-HCl buffer, pH 7.4; 20 mM KCl; 100 mM NaCl; 6 mM MgCl_2_; 3 mM ATP; 50–100 μg protein/sample. The final volume of the sample is 1 mL.

In order to inhibit the activity of Na^+^, K^+^-ATPase, 1 mM ouabain is added to the composition of the medium described above for quantification of the total ATPase activity. The final volume of the sample is 1 mL.

The tubes with the incubation medium together with the test substance (L-glutamate; kainic acid) at appropriate final concentrations (10^−12^, 10^−9^, 10^−6^, 10^−3^, 10^−2^ M) are incubated for 10 min at 37 °C.

The enzymatic reaction is initiated by the addition of ATP. After 15 min of incubation, the reaction is stopped by adding 1 mL of TCA (trichloroacetic acid), whereby the total volume of the samples becomes 2 mL. The samples are kept cold for 30 min, then centrifuged, whereupon the proteins precipitate and 0.5 mL of the supernatant is taken to determine the inorganic phosphate. Inorganic phosphorus (Pi) [43] and the protein content are determined [44].

The activity of Na^+^, K^+^-ATPase is defined as the difference between the activities of Na^+^, K^+^, Mg^2+^-ATPase (total ATPase) and Mg^2+^-ATPase.

Enzyme activities are defined as µM Pi/mg protein/hour.

### 2.5. Microelectrophoresis

Electrophoretic mobility studies are performed using a Cytopherometer (OPTON, Feintechnik Ges, mb. H., Wien, Austria). Electrophoretic migrations are measured for both forward and backward runs over a distance of 32 μm. The value of the EPM of synaptosomes is expressed in units of 10^−8^ m^2^V^−1^s^−1^. Values represent the mean of three independent preparations (52–119 synaptosomes). Only sharply visible single synaptic vesicles between 1–3 µm in diameter are measured. The zeta potential (ζ) is calculated from the electrophoretic mobility, *u*, using Helmholtz–Smoluchowski equation [45]:(1)$$\zeta =\frac{\eta u}{\varepsilon_{r}\varepsilon_{0}},$$
where ζ is in units of mV, $\varepsilon_{r}=78.5\;\left(at\;25\;{}^{\circ}C\right)$ is the relative dielectric permittivity of the aqueous phase, $\varepsilon_{0}=8.8542\times 10^{-12}\;Fm^{-1}$ is the permittivity of free space and η=0.001 Pa·s is the viscosity of the 50 mM Tris-HCl buffer (pH 7.2 at 25 °C), as in [46].

The electrostatic potential in the aqueous phase of the membrane surface at x.0 and charge density (σ) is given by:(2)$$\frac{A\sigma }{\surd C}=sinh\left(\frac{ze\Psi o}{2kT}\right),$$
where Ψo  is in mV, A=136.6  at 25 ℃,$\;A=1/\surd \left(8N\varepsilon_{r}\varepsilon_{0}kT\right)$, N=6022×10^23^ mol^−1^ is the Avogadro constant.

The charge density at the surface of the membrane reads [46]:(3)$$\frac{136.6\sigma }{\surd C}=sinh\left(\frac{z\Psi o}{51.38}\right),$$

The surface electrical charge is expressed in C·m^−h^. The values of surface charge density is almost certainly underestimated because of the assumption that ζ≅ Ψo  [46]. Using the buffer with low ionic strength ($I=0.050{\;M}^{-1}$) we minimize the above underestimation. Hence, ζ values in the present experiments are very close to the absolute value of $\Psi_{0}$ at the extracellular surface of the synaptosomal membrane.

### 2.6. Preparation of Giant Unilamellar Vesicles (GUVs)

The electroformation method [47] is applied for the preparation of GUVs from 1-palmitoyl-2-oleoyl-sn-glycero-3-phosphocholine (POPC, Avanti Polar Lipids Inc., Alabaster, AL 35007, USA) in 50 mM Tris-HCl buffer. The electrodes, representing two indium tin oxide (ITO)-coated glass plates, are separated by a polydimethylsiloxane (PDMS, Dow Corning, Midland, MI, USA) spacer [48]. A small amount (~50 μg) of POPC with concentration of 1 g/L in chloroform-methanol solvent (9:1 volume parts) is spread on the ITO-coated side of each of the electrodes. After the complete evaporation of the organic solvents under vacuum, the electroformation chamber is assembled in such a manner that the internal volume (~4 mL) is completely filled with 50 mM Tris-HCl buffer, pH7.2 for the control measurements or with 50 mM Tris-HCl buffer containing 0.01 mol/L of L-glutamic or kainic acid. All aqueous solutions are prepared with double-distilled water from a quarz distiller. Quasispherical unilamellar vesicles are obtained after the application of AC electric field to the chamber for a couple of hours, following previously established electroformation protocols [48,49].

### 2.7. Thermal Shape Fluctuation Analysis of GUVs

According to the theoretical model describing the mechanical properties of lipid bilayers [50,51], the elastic energy per unit area, *F_c_*, is given by the expression:(4)$$F_{c}=k_{c}{\left(c_{1}+c_{2}-c_{0}\right)}^{2}/2+k_{G}c_{1}c_{2}$$
where *k_c_* represents the measured quantity, namely, the bending modulus, *c*_1_ and *c*_2_ denote the principle modulus, *c*_0_ is the spontaneous curvature and *k_G_* stands for the Gaussian curvature modulus of the bilayer. The contribution of the latter vanishes in the case of GUVs’ closed membranes. In the present study we consider symmetrical bilayers with zero spontaneous curvature, $c_{0}=0\;$s. Thermal shape fluctuation analysis (TSFA) of giant lipid vesicles is applied to assess the bending modulus, *k_c_*, of lipid bilayers in the presence of 10 mM of the neurotransmitters studied [52,53,54,55,56].

The visualization of the membrane fluctuations is carried out by a phase-contrast inverted microscope (Axiovert 100, Zeiss, Germany) equipped with a two-channel thermocontrol (±0.1 °C) system operating both with the microscope plate and the oil-immersed objective (100×, numerical aperture 1.25). The image recording is performed by means of a CCD camera (C3582, Hamamatsu Photonics, Japan) and a frame grabber (DT3155, Data Translation, USA, 768 × 576 8-bit pixels, pixel size: 0.106 μm/pix) mounted in a Linux-operating station for digitization and analysis [53,54,55,56]. The stroboscopic illumination of the observed vesicles, synchronized with the camera, permitted the fast modes of fluctuations to be recorded and processed [55].

If the origin O of the frame of reference XYZ is placed in the center of the studied vesicle, and its axis Z is oriented parallel to the optical axis of the microscope, we can express the radius vector of a point at the vesicle surface at the moment t as $\rho \left(\theta ,\phi ,t\right)=R_{0}\left[1+u\left(\theta ,\phi ,t\right)\right]$ in the direction determined by its spherical coordinates (θ,φ). $R_{0}$ denotes the radius of a sphere with the same volume as the volume of the vesicle. The radius’ fluctuations here are represented by the normalized function u(θ,φ,t), which can be decomposed in a series of spherical functions [53]. Considering all modes as independent, the mean square value of the fluctuations has been obtained to depend on the number n only [57]: $\langle {\left|U_{n}^{m}\left(t\right)\right|}^{2}\rangle =k_{B}T/\left\{k_{c}\left(n-1\right)\left(n+2\right)\left[\bar{\sigma }+n\left(n+1\right)\right]\right\}$, where, $k_{B}$ stands for the Boltzmann constant, T is the absolute temperature, and $\bar{\sigma }=\sigma R^{2}/k_{c}$ denotes the dimensionless membrane tension, reflecting the vesicle excess area compared to the area of a sphere with the same volume as the volume of the vesicle. Hence, the product$\;\langle {\left|U_{n}^{m}\left(t\right)\right|}^{2}\rangle \cdot \left(n-1\right)\left(n+2\right)\left[\bar{\sigma }+n\left(n+1\right)\right]=k_{B}T/k_{c}$ is independent of n and $\bar{\sigma }$. It is used for the determination of the very small and otherwise immeasurable membrane tension $\bar{\sigma }\;$via Legendre analysis of the autocorrelation function of the vesicle contour [53,58].

In phase contrast, we are able to observe and analyse the equatorial cross-section of the vesicle in the focal plane of the microscope (Figure 1) for θ=π/2, thus obtaining$\;\rho \left(\phi ,t\right)=R_{0}\left[1+u\left(\frac{\pi }{2},\phi ,t\right)\right]$. In order to perform static fluctuation analysis, we calculate the normalized angular autocorrelation function ξ(γ,t) of the vesicle radius at a given moment of time t [58], namely, $\xi \left(\gamma ,t\right)=\langle \frac{1}{R_{0}^{2}}\left[\int_{0}^{2\pi }\rho \left(\phi +\gamma ,t\right)\rho^{*}\left(\phi ,t\right)d\phi -\rho^{2}\left(t\right)\right]\rangle$.

The time averaged autocorrelation function ξ(γ) is represented as a series of Legendre polynomials, $P_{n}\left(\cos\gamma \right)$, with coefficients $B_{n}\left(\bar{\sigma },k_{c}\right)=\frac{k_{B}T}{4\pi k_{c}}\frac{\left(2n+1\right)}{\left(n-1\right)\left(n+2\right)\left[\bar{\sigma }+n\left(n+1\right)\right]},n\geq 2\;$ [53,58], establishing a direct relation between the experimentally measurable vesicle radius ρ(φ,t) and the amplitudes of the vesicle fluctuations, reading $B_{n}\left(\bar{\sigma },k_{c}\right)=f_{n}^{corr}\frac{\left(2n+1\right)}{4\pi }\langle {\left|U_{n}^{m}\left(t\right)\right|}^{2}\rangle$. Here $f_{n}^{corr}$ stands for a multiplicative correction factor accounting for the finite time of acquisition of every image due to the finite integration time $t_{s}$ of the CCD camera used [53]. In the experimental set up applied here, $t_{s}=$ 40 ms.

The relaxation of the fluctuating membrane of a quasispherical GUV [57] is related to the vesicle volume, the surrounding medium viscosity η and the bilayer bending elasticity, as well as to the vesicle excess area, which is the difference between the area of the studied vesicle and the surface area of a sphere with the same volume as the volume of the vesicle. The correlation time calculated for the amplitude $U_{n}^{m}\left(t\right)$ reads [57] $\tau_{n}^{m}=\frac{\eta R_{0}^{3}}{k_{c}}\frac{2n+1}{\left(n-1\right)\left(n+2\right)\left[\bar{\sigma }+n\left(n+1\right)\right]}\left(2-\frac{1}{n\left(n+1\right)}\right)$.

In the present study, we take into consideration only vesicles with correlation time of the slowest fluctuation mode (n= 2), $\tau_{2}^{m}\;\leq$ 0.5 s. As far as we analyze image sequences with the time lapse between adjacent frames of 1 s, only snapshots of non-correlated shape fluctuations are used in the calculation of the membrane bending modulus.

The fluctuation analysis is performed up to n= 19 and, thus, the shortest correlation time is estimated at $\tau_{19}^{m}\approx$ 6 ms. In our experimental set up, vesicles are stroboscopically illuminated by light pulses (shorter than 4 μs [59]). Accordingly, no correction factor is applicable ($f_{n}^{corr}=1$) in the calculation of $B_{n}\left(k_{c},\bar{\sigma }\right)$ coefficients.

The contour representing the equatorial cross section of the vesicle membrane is extracted using a procedure reported in detail elsewhere [53,56]. The fitting procedure for obtaining $k_{c}$ and $\bar{\sigma }$ values is performed according to [53,56]. While $\bar{\sigma }$ varies for different vesicles, the value of the material constant $k_{c}$ is considered the same for the whole ensemble of vesicles in each sample.

Only visibly fluctuating quasispherical vesicles larger than 10 μm in diameter are chosen for TSFA. Several hundred images (at least 400) are captured once per second and subjected to TSFA for calculation of the membrane bending constant and tension of every vesicle studied [53,54,55]. The reported values of the membrane bending modulus are obtained over an ensemble of not less than five GUVs collected from at least three different electroformation batches. All vesicles used for the calculation of $k_{c}$ satisfy the criteria for quality previously established [54,56], including the absence of defects and/or heterogeneities in the membrane; preservation of vesicle volume during measurements; uniformity of the mean radius of the vesicular contour over its all angular directions; and sequences of non-correlated images taken for analysis for every studied vesicle.

All measurements of the bending elasticity are carried out at (37 ± 0.1) °C.

### 2.8. Statistical Analysis

The electrokinetic data are averaged as triplicate measurements for every sample. The data are expressed as mean ± SE. The significant means are determined by use of ANOVA. One-way analysis of variance is performed with Dunn’s Test following ranked–based ANOVA (Kruskal–Wallis one way analysis of variance on ranks) and Student–Newman–Keuls method, taking *p* < 0.05 as significant, *p* < 0.01 as highly significant and *p* < 0.001 as extremely significant and represented by an asterisk in the figures.

The experimental data of ATPase assay are analyzed by Student’s *t*-test. Differences from controls are considered significant at *p* < 0.05.

## 3. Results

The studies are performed with a semi-purified synaptosomal fraction isolated from the cerebral cortex of rat brain [8]. This is the most commonly used object in such scientific research. The concentrations of the substances used range from 10^−12^ M to 10^−2^ M. Hence, physiological concentrations are covered together with higher and lower concentration ranges.

### 3.1. Effect of L-Glutamate and Kainic Acid on the Adenosine Triphosphatase Activities on Synaptic Membranes (ATPase Assay)

The results of the experimental studies are shown in Figure 2. L-Glutamate in final concentrations from 10^−12^ M to 10^−6^ M does not affect Na^+^, K^+^-ATPase. At concentrations of 10^−3^ M and 10^−2^ M Glu (the second higher than the average physiological concentration), a significant increase in enzyme activity is observed, respectively, 16% and 42% above the value of the control samples (Figure 2a).

Glutamate has a positive nonspecific effect on Mg^2+^-ATPase at all the concentrations of treatment. Enzyme activity increases within 17% to 24% of control. The same nonspecific increase in activity is observed in the experiments with total ATPase (Figure 2a).

According to the obtained results for the influence of KA in final concentrations from 10^−12^ M to 10^−2^ M on Na^+^, K^+^-ATPase (Figure 2b), the substance increases the enzyme activity at concentrations from 10^−6^ M to 10^−2^ M with ~20% to 40%.

Mg^2+^-ATPase is not affected by kainic acid (Figure 2b). The increase in the activity of total ATPase is due to the increased Na^+^, K^+^-ATPase and occurs at 10^−3^ M and 10^−2^ M concentrations of KA (Figure 2b).

### 3.2. Influence of L-Glutamate and Kainic Acid on the Enzymatic Activity of Brain Acetylcholinesterase (AChE)

Assay of acetylcholinesterase (AChE) activity is important in in vitro characterization of L-Glu and KA as amino acids and neurotransmitters during treatments of synaptic membranes.

The final concentrations of L-glutamate from 10^−12^ M to 10^−2^ M are used. At concentrations from 10^−12^ M to 10^−3^ M, no significant effect of AChE is found (Figure 3a). At 10^−2^ M L-Glu, a nearly 30% decrease in activity is observed compared to control samples (Figure 3a).

Figure 3b presents the results for the influence of the glutamate agonist—KA on ATPase activity of synaptic membranes. Its effect is similar to that of L-Glutamate. We report the preservation of the acetylcholinesterase activity of synaptic membranes in the presence of 10^−12^ M to 10^−3^ M kainic acid. At higher concentrations of KA (10^−2^ M), we measure an enzyme inhibition of nearly 30% compared to the control sample.

### 3.3. Influence of L-Glutamate and Kainic Acid on the Electrokinetic Parameters of Synaptosomes

One of the parameters that reflects the physicochemical state of synaptosomes is their surface electric charge. When it interacts with the ions in the aqueous solution, a double electric layer is formed. This layer determines the local electric field in the solution layer adjacent to the vesicular surface. The surface charge plays an important role in synaptosome–synaptosomal interactions (and predetermines the concentration profile of various biological active compounds) including penetrating ions and charged substrates for the corresponding synaptosomal receptors located in close proximity to the membrane.

The dynamics of surface electrical charge on the membrane is given by the parameter electrophoretic mobility (EPM). An increase in the EPM of the synaptosomes (1–4 h after their isolation) is observed in the presence of 10^−9^–10^−3^ M Glu in the suspending medium (Figure 4a). The observed effect is associated with an increase in the zeta potential in an appropriate concentration range.

A concentration of 10^−3^ M Glu significantly affects EPM at 17–22 h after the isolation of synaptosomes (Figure 4a) where zeta potential decreases (Figure 5a). Synaptosomes observed on the second day of their isolation are characterized by complexes with 3–5 synaptosomes in them. There is a slight increase in the EPM of the synaptosomes in the presence of 10^−9^ Glu, which does not significantly affect the surface electric charge of the synaptosomes on the third day after their isolation (Figure 4a and Figure 6a). A dose of 10^−6^ M Glu causes the formation of a large aggregate of synaptosomes on the second day of their isolation. In the presence of 10^−9^ M Glu, slightly swollen synaptosomes are observed on the second day of isolation, some of them adhered in a structured aggregate.

Kainic acid leads to a strong effect of EPM on synaptosomes 1–4 h after their isolation in the concentration range of 10^−12^–10^−3^ M KA (Figure 4b). This effect is associated with a strong increase in electrokinetic potential (Figure 5b) and surface charge (Figure 6b) during treatment with 10^−12^ M, 10^−9^ M or 10^−3^ M KA on the synaptosomal membrane.

EPMs of synaptosomes 17–22 h after their isolation are characterized by an increase in EPM in the presence of 10^−12^ M KA (Figure 4b) and an increase in zeta potential by 2.7 mV (Figure 5b). A decrease in the EPM of the synaptosomes is observed on the second day after their isolation after exposure by 10^−6^ and 10^−3^ M KA, associated with a decrease in ζ, but without changes in σ at all treatment concentrations used (Figure 6b).

During in vitro aging, 10^−12^ M KA leads to a sharp decrease in EPM (Figure 4b) and the zeta potential of synaptosomes at 41–47 h after their isolation, where ζ decreases by 3.2 mV compare to control values (Figure 5b). No changes in the surface electric charge of the synaptosomes are recorded on the third day after their isolation in the entire concentration range of treatment (Figure 6b).

At pH 7.2, the measured electrokinetic potential of the synaptosomes immediately after their isolation in 1–4 h is ζ = −19.0 mV. At 17–22 h after isolation, the synaptosomes retain almost the same value of the negative zeta potential of ζ = −21.1 mV, as in the zeta potential characterizing the synaptosomes at 41–47 h after isolation of ζ = −21.0 mV. The observed increase in EPM in the in vitro aging process of synaptosomes can be explained by the exposure of ionic groups from the surface or the occurrence of dynamic defects in the membrane (Figure 4).

The surface charge density of synaptosomes during in vitro aging increases from σ = −0.0099 C m^−2^ immediately after their isolation, as well as at 17–22 h (σ = −0.0113 C m^−2^) to −0.0110 C m^−2^ in 41–47 h. This represents a 13.7% enhancement in the surface electric charge of the synaptosomes during in vitro aging (Figure 6).

In 1–4 h of the in vitro aging of synaptosomes incubated with L-glutamate (10^−3^ M), a maximum effect of a ~15% increase in the negative density of the surface charge is observed (Figure 6a). At 17–22 and 41–47 h after isolation of synaptosomes, no statistically significant changes in their surface electric charge are registered excepting the case of 10^−3^ M Glu treatment where a small decrease in σ (*p* = 0.048) is observed (Figure 6a). The formation of more swollen synaptosomes is observed on the third day of their measurement, 3–5 in aggregate, in the presence of 10^−9^ M Glu. In the presence of 10^−6^ M Glu, a slight reorientation of the synaptosomes during movement to the cathode is observed in the presence of DC. In the presence of 1 mM Glu, the synaptosomes are more swollen than those of the control, untreated synaptosomes. Most synaptosomes form aggregates, with some of the vesicles changing their place in the aggregates during their movement in an electric field.

Kainic acid-treated (10^−12^ M, 10^−9^ M, 10^−3^ M) synaptosomes significantly change the surface electrical charge in 1–4 h after their isolation. The aggregation of the synaptosomes (aggregates of two in a complex) is observed, as well as those arranged in structured aggregates in the presence of 10^−12^ M KA after their isolation. In the presence of 10^−9^ M KA, the reorientation of the synaptosomes is seen immediately after their isolation in the complex, as well as more swollen synaptosomes than those in the control. Furthermore, 10^−6^ M KA causes the reorientation of some of the complexes in the course of their movement during the microelectrophoretic measurement in 1–4 h after their isolation.

In 17–22 h of the in vitro aging of synaptosomes incubated with 10^−3^ M KA, a 16% effect of increasing EPM is observed, and at 41–47 h no alteration of the negative charge density is registered (Figure 6b). Net surface charge effects range from 13% in the presence of 10^−12^ M KA in the suspension medium to 17% in the presence of 10^−9^ M KA (Figure 6b). There are good structuring aggregates in the presence of 10^−12^ M KA and at 10^−9^ M KA (aggregates of 4–5 synaptosomes in a complex) during the second day of synaptosome isolation. In the presence of 10^−6^ M KA, the synaptosomes on the second day of isolation appear to be polarized and reorient in the course of the electric field. There are various complexes involving 3–4 synaptosomes during microelectrophoretic measurement before and after changing the direction of electric current. The higher concentration of 10^−3^ M KA causes ordered aggregates and complexes of synaptosomes that are more swollen than those observed at 10^−9^ M KA on the second day of isolation.

More swollen synaptosomes and structured synaptosome complexes are observed on the third day of their isolation in the presence of 10^−12^ M KA. In addition, 10^−9^ M KA leads to the formation of synaptosomal aggregates, several in one complex. 10^−6^ M KA induces structured aggregates, some of the synaptosomes on the third day of isolation move at a very low rate compared to the single synaptosomes measured during the experiment. Furthermore, 10^−3^ M KA leads to reorientation of synaptosomes in the course of movement in an electric field on the third day of their isolation. The reorientation of the whole synaptosome complex and of individual synaptosomes in it is observed in the course of microelectrophoretic measurement without changing the direction of current flow.

### 3.4. Bending Elasticity of Model Lipid Membranes in the Presence of L-Glutamate and Kainic Acid

The control measurements of POPC membrane bending rigidity are performed both in bidistilled water and in 0.05 M Tris-HCl buffer. The obtained results indicate that the bending modulus is not significantly altered in the buffer solution compared to its value in water (Table 1 and Figure 7). The electroformation in 10 mmol/L of L-Glu yields small vesicles and foam-like adhered membrane structures non-appropriate for TSFA (see Appendix A). Hence, we prepare GUVs in buffer containing ten—fold lower content of the neurotransmitter, namely, 1 mM L-Glu. In the latter milieu the electroformation protocol produces well fluctuating quasispherical vesicles, allowing for shape fluctuation analysis and the assessment of the membrane bending modulus.

The calculated $k_{c}$ and $\bar{\sigma }$ values, with their goodness of fit for all recorded and analyzed GUVs, can be found in SM. For the four GUVs populations, including two controls and two buffer solutions of L-Glu or KA (Table 1), we calculate the best data fit by $\chi^{2}$ minimization [53].

The bending modulus of POPC bilayers modified by 1 mmol/L of L-Glu decreases by nearly ~25% compared to its value obtained for bare POPC bilayers. This result testifies for the L-Glu softening effect exerted on lipid membranes at the concentration of 1 mmol/L studied. In the presence of a 10-fold higher concentration of KA, no statistically significant alteration of the membrane bending elasticity is obtained, as shown in Table 1 and Figure 7.

## 4. Discussion

The complex functional organization of the CNS balances the interaction between the two main processes of excitation and inhibition, which are highly dynamic processes. Both excitation and inhibition occur due to the involvement of neurotransmitter systems in the transmitted information at the synapse from neuron to neuron or effector cell. The brain functional networks are altered and coupled to neuropathology and cognitive decline [60].

For the nerve cell membrane, Na^+^, K^+^-ATPase is the enzyme without which it is impossible to maintain the stability of ion gradients and membrane potential [61]. Numerous ion pumps in the neural membrane, which create and maintain a critical level of membrane potential and ion gradients across the membrane, use ATP as an energy source, mainly of mitochondrial origin. Most of the ATP that is produced in the brain is used to restore the ionic gradients that have been altered as a result of synaptic transmission. Phosphatidylserine and phosphatidylglycerol as negatively charged lipids promote the bilayer formation of physiological thickness and increased fluidity tends to support optimal Na^+^, K^+^-ATPases activity [62]. Na^+^, K^+^-ATPases have been involved in cell signaling, interacting with partner proteins [63].

The depolarizing stimuli caused by L-Glutamate, especially at higher concentrations of the latter, logically need a more effective rate of recovery of the ionic status quo, i.e., repolarization should be performed with the help of an activated Na^+^, K^+^-ATPase with greater repolarizing potential. Stimulation of Na^+^, K^+^-ATPase by L-Glutamate increases the effectiveness of the sodium–calcium antiport system, which normally removes Ca^2+^ from the cell. Our results show that L-Glutamate has a positive nonspecific effect on Mg^2+^-ATPase. Cholinesterase activity is a biomarker of neurotoxicity [64]. The interest is provoked, on the one hand, by the established involvement of AChE in the inactivation of AChE in a large part of the synapses in the CNS and neuromuscular synapses. On the other hand, practical interest is generated by the easy vulnerability of AChE to chemicals, some of which are used as war poisons and others as pesticides in agriculture.

The presented results show that 10 mmol/L of L-glutamate and kainic acid inhibit AChE activity, while lower concentrations of the neurotransmitter/agonist studied do not exert any measurable effect on enzyme activity. The inhibition is more expressed in the presence of kainic acid and affects not only the metabolic enzymes inside the cell, but also the membrane components, such as the integral membrane protein AChE.

We report that lower concentrations of L-glutamate do not alter Na^+^, K^+^-ATPase and AChE activity. The lack of effect at small concentrations of the effectory neurotransmitter on Na^+^, K^+^-ATPase and AChE is a part of the normal cellular metabolism. If L-Glu did affect the activity of enzymes at normal concentrations, this would lead to the disruption of the processes it controls. Alterations can be observed after the nerve impulse accompanied by local transitory changes of the mediator concentration. We confirm the stimulating effect of L-glutamate at concentrations of 10^−3^ M and 10^−2^ M on Na^+^, K^+^-and Mg^2+^-ATPases enzymatic activity in vitro concerning the activation of the enzymatic system responsible for active transport of cations across synaptic membranes from the cerebral cortex. The presented results support that KA enhances the enzyme activity of the Na^+^, K^+^-ATPase of in vitro studies of synaptic membranes, where the activation of cation transport across the membrane is expected.

The energy of the putative interactions between synaptosomes can be expressed approximately as a function of geometric electrostatic and London parameters on the surface of the intermediate phase [45,65]. It is believed that the surface electric charge, measured by microelectrophoresis, is closely related to the intracellular processes in neuronal cell [66]. In order to provide information about the biochemical microenvironment of the synaptosomal surface, we applied microelectrophoresis for the characterization of synaptosomes’ electrokinetic properties after exposure to L-glutamic and kainic acids. The analysis of the acquired data allows for assessing the zeta potential as a marker for the stability of synaptosomes in particle electrophoresis.

In the presence of L-glutamic and kainic acids, the surface electrical charge of synaptosomes from the cerebral cortex shows a moderate increase. This effect reaches its maximum on the first day after the isolation of synaptosomes upon treatment with L-glutamate and kainic acid in vitro.

Microelectrophoretic studies have shown that L-Glu affects the electrophoretic mobility of synaptosomes and increases their negative charge density at treatment concentrations of 10^−9^ to 10^−3^ M, as a result of exposing additional negatively charged groups to the surface of synaptosomes immediately after isolation. KA, as a potent agonist, increases the negative charge of synaptosomes to 1–4 h after isolation in the entire concentration range of treatment, from 10^−12^–10^−3^ M, which emphasizes its more effective action on the surface properties of synaptosomal membranes. The upper electrokinetic effects of synaptosomes are electrostatic in nature with no membrane permeability alterations in the presence of glutamate/kainic acid observed. We assume that the effects of increasing the surface charge under the influence of L-Glutamate and KA may be accompanied by an increase in the free radical products of the biological system [67], which is in the scope of further research.

Electrokinetic data for synaptosomes acquired here provide information about the dependence of their EPM and surface electrical charge on L-Glu and KA concentrations in the buffer. Our previous results of the EPM of synaptosomes isolated from cerebral cortex, suspended in a saline–sorbitol buffer with low ionic strength in the presence of L-Glu and KA, have shown strong volume changes in synaptosomes in the presence of the neurotransmitters studied [11]. There are numerous phenomenological data that seek to elucidate the coupling mechanism between hypothetical membrane sensors responding to changes in volume and executive transport systems, but, in practice, this mechanism is not fully clarified. One of the possible pathways for volume regulation is associated with the metastable structural organization of the membrane. This stressful state is created by the interaction of the protein cytoskeleton and lipid bilayer under the action of electric or osmotic potentials, which leads to modifications in the activities of a number of transport systems [68]. It is worth noting that the elasticity of neurons increases abruptly with hypotonicity, where neurons invariably softened towards or below the pretreatment level [6]. In the present study, we establish a relationship between the interaction of neurotransmitters with model phosphatidylcholine membranes and the bending elasticity of the bilayer. The results obtained on model lipid bilayers support the conclusion that L-Glu induces a softening effect on the lipid matrix, while KA does not significantly change the bending modulus of the bilayer. The decrease in the membrane bending stiffness in the presence of L-Glu relates to the interaction of lipid bilayers with glutamate anions at pH 7.22 [69]. L-Glutamic acid, which is characterized by a linear structure and lower molecular weight compared to KA, interacts more readily with the headgroup region of the bilayer, thus leading to a measurable decrease in its bending modulus [70,71]. Based on the data acquired on model lipid membranes, their bending rigidity is considered as a mechanical parameter, which is sensitive to the presence of L-Glu in the suspending buffer and remains unchanged in a KA containing aqueous phase.

L-Glu and KA alter synaptosome behavior in particle electrophoresis immediately after treatment. In the presence of L-Glu, synaptosomes become more negatively charged. Hence, their electrochemical stability increases in relation to the NT importance in mental health and memory [7]. The increase in synaptosome net charge at all doses of KA-treatment enhances the membrane electrokinetic stability without the alteration of the membrane rigidity. This fact might be of importance in the elucidation of additional mechanisms of KA action in the evocation of status epilepticus [72]. Taking into account the established modulation of enzymatic activity by exerting a mechanical stress on enzymes [73], we consider the bending modulus as an important physical parameter to be investigated with regard to the membrane-neurotransmitters interactions.

## 5. Conclusions

The results from microelectrophoretic studies obtained on isolated synaptosomes from the cerebral cortex during in vitro aging consider circumstances that occur in the compartment in synapses in vivo. An increase in the surface electrical charge of synaptosomes treated with L-Glu or KA during in vitro aging is reported, which is related to the specific balance of their impact on processes affecting surface properties of synaptosomes. Higher local surface charge densities induce membrane association [74] by acting directly on the enzyme activity, which could also include a change in the membrane mechanical parameters upon the action of the studied neurotransmitter and its agonist.

As far as acetylcholinesterase activity is used as a parameter for the biomonitoring of pollution or environmental damage [75], a significant result of our study is that glutamic and kainic acids reduce acetylcholinesterase activity at 10 mM. The decreased AChE activity in synaptosomal fraction from the cerebral cortex upon L-Glu and KA treatment in vitro may cause disturbances of the ionic balance in synaptic membranes and, thus, deviations in the normal functions of synapses related to the discrete character of nerve transmission.

The acquired results manifest the modulation of the electrokinetic properties of synaptosomes and enzyme activity of synaptic membranes from the rat brain cerebral cortex upon the action of signaling molecules such as glutamate and kainate. We provide evidence of the softening effect exerted by the excitatory neurotransmitter on the mechanical properties of the lipid matrix. The reported data is expected to shed light on the path towards further revealing the molecular mechanisms of nerve cell activity. The latter is considered as a prerequisite for fostering the development of novel therapeutic approaches in the treatment of neurodegenerative conditions.

## Figures and Tables

**Figure 1 membranes-11-00987-f001:**
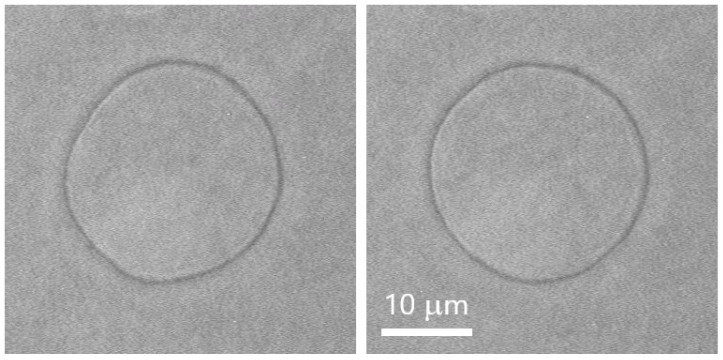
Phase contrast micrographs of POPC vesicle with radius 11.8 µm in 1 mM L-Glu, 50 mM Tris-HCl, pH 7.2; two subsequent images captured at 1 frame per second; scale bar corresponds to 10 µm.

**Figure 2 membranes-11-00987-f002:**
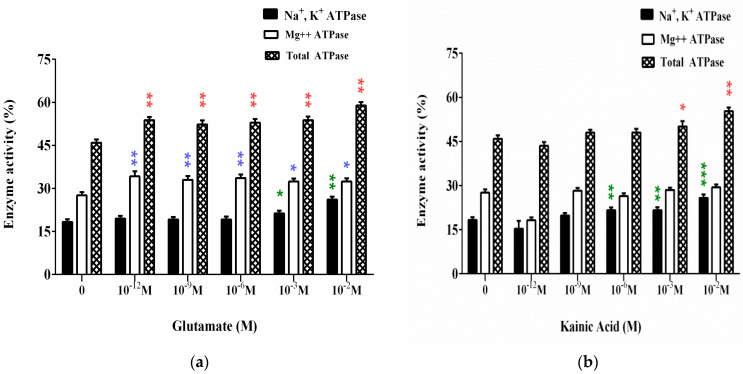
Effect of L-Glutamate (**a**) and kainic acid (**b**) at different concentrations on Na^+^, K^+^-ATPase (green color), Mg^2+^-ATPase (blue color) and total ATPas (red color) of synaptic membranes from the cerebral cortex. Here, 100% enzyme activity corresponds to: for Na^+^, K^+^-ATPase—18.3 μmol inorganic phosphate/mg protein/hour; for Mg^2+^-ATPase—27.6 μmol inorganic phosphate/mg protein/hour; for total ATPase—45.9 μmol inorganic phosphate/mg protein/hour. SE, as determined from four separate experiments, each assayed in triplicate (standard error of number of measurements *n* = 12 at * *p* < 0.05; ** *p* < 0.01; *** *p* < 0.001).

**Figure 3 membranes-11-00987-f003:**
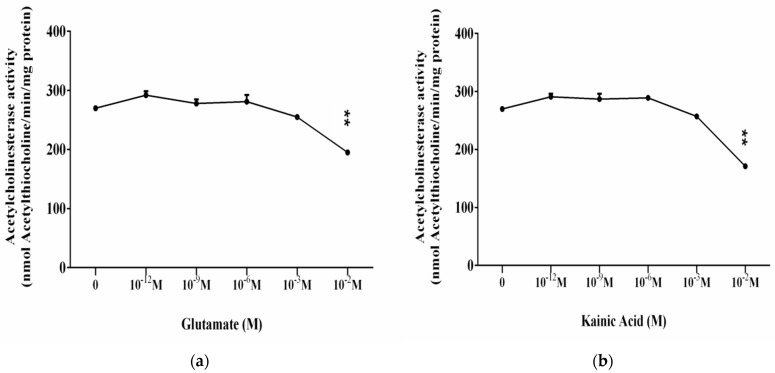
Effect of L-Glutamate (**a**) and kainic acid (**b**) on the acetylcholinesterase activity of synaptic membranes from the cerebral cortex. Effect of L-glutamate/kainic acid at concentrations from 10^−12^ M to 10^−2^ M on the enzymatic activity of AChE of synaptic membranes/nmol degraded acetylthiocholine/min/mg protein. SE, as determined from four–five separate experiments, each assayed in triplicate (standard error of *n* = 12–18 at ** *p* < 0.01).

**Figure 4 membranes-11-00987-f004:**
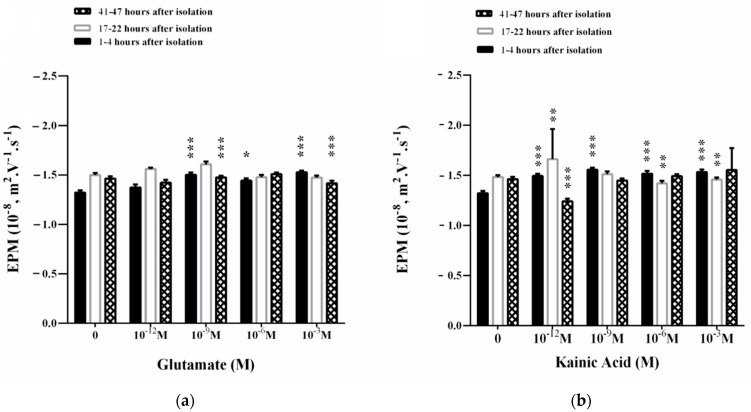
Electrophoretic mobility (EPM) of synaptosomes from the cerebral cortex after treatment with L-glutamate (**a**) or kainic acid (**b**). Synaptosomes are treated for 30 min at 37 °C. Electrophoretic mobility measurements are performed at 25 ± 0.1 °C in 0.05 M Tris-HCl buffer, pH 7.2. Each value is the mean ± SE of three independent preparations. * *p* < 0.05; ** *p* < 0.01; *** *p* < 0.001.

**Figure 5 membranes-11-00987-f005:**
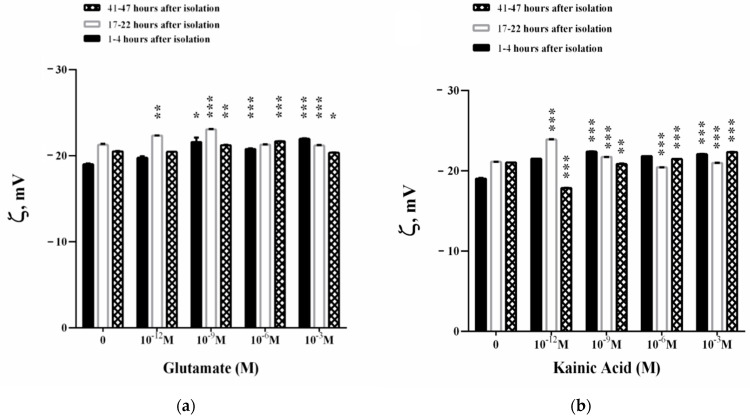
Zeta potential (ζ) of synaptosomes from the cerebral cortex after treatment with (**a**) L-glutamate and (**b**) kainic acid. Synaptosomes are treated for 30 min at 37 °C. Zeta potential measurements are performed at 25 ± 0.1 °C in 0.05 M Tris-HCl buffer, pH 7.2. Each value is the mean ± SE of three independent preparations. * *p* < 0.05; ** *p* < 0.01; *** *p* < 0.001.

**Figure 6 membranes-11-00987-f006:**
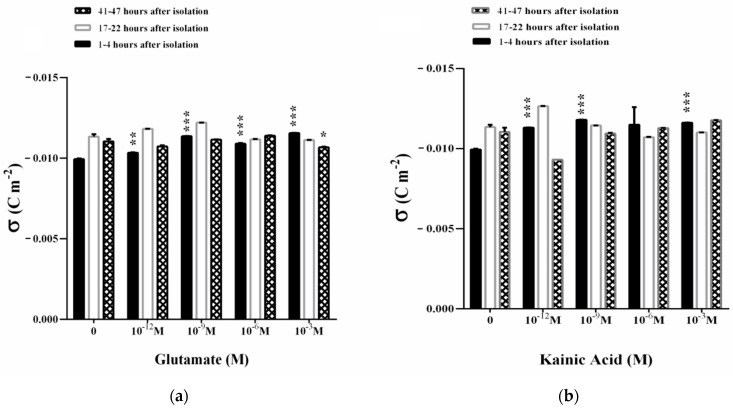
Surface electrical charge (σ) of synaptosomes from the cerebral cortex after treatment with L-glutamate (**a**) or kainic acid (**b**). Synaptosomes are treated for 30 min at 37 °C. Surface electrical charge measurements are performed at 25 ± 0.1 °C in 0.05 M Tris-HCl buffer, pH 7.2. Each value is the mean ± SE of three independent preparations. * *p* < 0.05; ** *p* < 0.01; *** *p* < 0.001.

**Figure 7 membranes-11-00987-f007:**
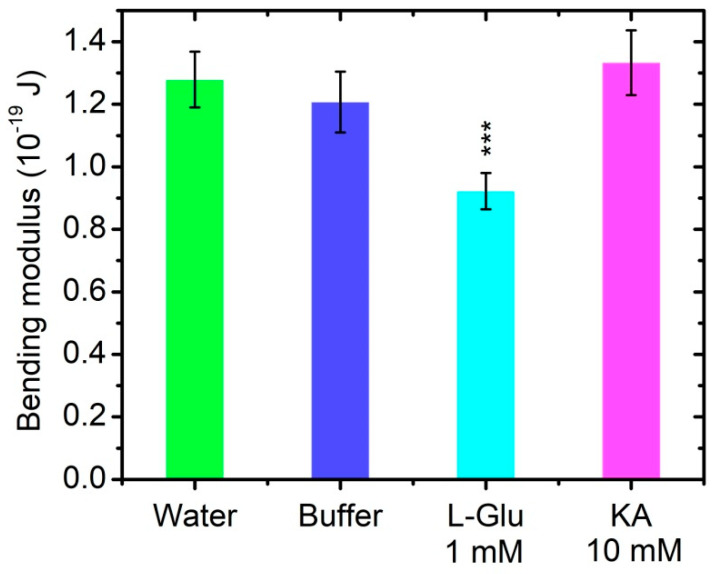
Bending modulus of POPC bilayers in 0.05 M Tris-HCl buffer, pH 7.2 in the presence of 1 mmol/L L-Glu (*** *p* < 0.001) or 10 mmol/L KA (n.s.) at (37 ± 0.1) °C. The bending modulus is presented as weighed mean ± SD.

**Table 1 membranes-11-00987-t001:** Bending elasticity of POPC membranes and the effect of the neurotransmitters (NT) L-Glu and KA; data acquired from TSFA of GUVs at (37 ± 0.1) °C control sets of measurements performed in: (i) bidistilled water and (ii) 50 mM Tris-HCl buffer, pH 7.2; GF—goodness of fit ($\chi^{2}$-test).

Sample	NT, mmol/L	Bending Modulus, 10^−en^ J	Number of Vesicles	GF
(i) Control, H_2_O	0	1.28 ± 0.09	17	0.59
(ii) Control, buffer	0	1.21 ± 0.10	8	0.31
L-Glutamate	1	0.92 ± 0.06	9	0.62
Kainic acid	10	1.33 ± 0.10	5	0.77

## Data Availability

Data is contained within the article and Appendix A.

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
