# Peer review of "Surface Properties of Synaptosomes in the Presence of L-Glutamic and Kainic Acids: In Vitro Alteration of the ATPase and Acetylcholinesterase Activities"

_membranes, 2021, doi:10.3390/membranes11120987_

Round 1
Reviewer 1 Report
Title: “ Surface properties of synaptosomes in the presence of L-glutamic and kainic acids: in vitro alteration of the ATPase and acetylcholinesterase activities”
In this work the authors measured decrease in the bending modulus of model bimolecular membranes composed of monounsaturated lipid 1-palmitoyl-2-oleoyl-sn-glycero-3-phosphocholine provides evidence for softer membranes in the presence of L-glutamate. The authors claim that kainic acid does not affect the membrane mechanical stability even at ten-fold higher concentrations. The authors presumed that both the L-glutamic and kainic acids reduce acetylcholinesterase activity and deviation from normal functions of neurotransmission in synapses. Finally, the authors claim that the presented results regarding the modulation of the enzyme activity of synaptic membranes and surface properties of synaptosomes are expected to contribute in elucidation of the molecular mechanisms of neurotransmitters/agonists’ action on membranes.
General comment: Although the aim of this work seems to be interesting, the manuscript should be revised to enhance its quality and impact. In particular, some more analysis should be performed in order to better support some main claims of the authors, while some additional minor issues should be reworked within the “Results” section.
Some detailed comments:
*) Within the “Results” section some scattered lines of “Methods” are present. They should be moved to the “Methods” section.
Some examples:
lines: “3.2. Influence of L-glutamate and kainic acid on the enzymatic activity of brain acetylcholinesterase
346 (AChE)
347 Assay of acethylcholinesterase (AChE) activity is important in in vitro characteriza-
348 tion of L-Glu and KA as amino acids and neurotransmitters during treatments of synap-
349 tic membranes.
350 The final concentrations of L-glutamate from 10-12 M to 10-2 M are used. “
lines: “3.3. Influence of L-glutamate and kainic acid on the electrokinetic parameters of synaptosomes
364 One of the parameters that reflect the physicochemical state of synaptosomes is their
365 surface electric charge. When it interacts with the ions in the aqueous solution, a double
366 electric layer is formed. This layer determines the local electric field in the solution layer
367 adjacent to the vesicular surface. The surface charge plays an important role in synapto-
368 some - synaptosomal interactions (and predetermines the concentration profile of various
369 biological active compounds) including penetrating ions and charged substrates for the
370 corresponding synaptosomal receptors located in close proximity to the membrane. “
lines: “3.4. Bending elasticity of model lipid membranes in the presence of L-glutamate and kainic acid
467 The electroformation in 10 mmol/L of L-Glu yields small vesicles and foam-like
468 structures non-appropriate for TSFA. Hence, we prepare GUVs in buffer containing
469 1 mM L-Glu. In the latter milieu the electroformation protocol produces well fluctuating
470 quasispherical vesicles, allowing for shape fluctuation analysis and deduction of the
471 membrane bending modulus. “
etc
lines: “472 The bending modulus of POPC bilayers modified by 1 mmol/L of L-Glu decreases
473 by nearly ~25 % compared to its value obtained for bare POPC bilayers. This result
474 supports the hypothesis about L-Glu softening effect exerted on lipid membranes at the
475 concentration of 1 mmol/L studied. In the presence of a 10-fold higher concentration of
476 KA no statistically significant alteration of the membrane bending elasticity is obtained
477 as shown in Table 1 and Figure 6.
Table 1. Bending elasticity of POPC membranes and the effect of the neurotransmitters (NT) L-Glu
479 and KA; data acquired from TSFA of GUVs at (37 ± 0.1) ℃; control measurements performed in
480 50 mM Tris-HCl buffer, pH 7.2.
Figure 6. Bending modulus of POPC bilayers in 50 mM Tris-HCl buffer, pH 7.2 in the presence of
484 1 mmol/L L-Glu (*** p < 0.001) or 10 mmol/L KA (n.s.) at (37±0.1) ℃. The bending modulus is pre-
485 sented as mean SD.
*) The authors should better show in a quantitative way that the found difference in bending modulus is statistically significant and, in particular, that this change can lead to different biological behaviours and therefore is biologically relevant. This is a crucial point for this work.
Lines: “mechanical parameters upon the action of the studied neurotransmitter and its agonist.
612 The acquired results manifest the modulation of the electrokinetics of synaptosomes
613 and enzyme activity of synaptic membranes from the rat brain cerebral cortex upon the
614 action of signaling molecules such as glutamate and kainate. We provide evidence of the
615 softening effect exerted by the excitatory neurotransmitter on the mechanical properties
616 of the lipid matrix. The reported data is expected to shed light on the path towards fur-
617 ther revealing the molecular mechanisms of nerve cell activity. The latter is considered as
618 a prerequisite for fostering the development of novel therapeutic approaches in the
619 treatment of neurodegenerative conditions. “
*) These claims should be better supported by quantitatively showing the statistical significance of the found change of bending stiffness of membranes.
Author Response
Dear Reviewer,
Please find enclosed the revised manuscript Membranes-1483104 and Response to reviewer.
Sincerely yours,
Virjinia Doltchinkova

Reviewer 2 Report
The manuscript has been revised well about my suggestions. I think this manuscript will be adequate for acceptance.
Author Response
Dear Reviewer,
Plese find enclosed the revised manuscript Membranes-1483104 and Response to Reviewer.
Sincerely yours,
Virjinia Doltchinkova

Reviewer 3 Report
In this paper, the authors investigated the effect of L-glutamic and kainic acids on ATPases and acetylcholinesterase in synaptic membranes prepared from cerebral cortex of rat brain tissue. The experiments are well carried out and the manuscript is well organized. Few concerns regarding the study are pointed out below:
- Number of rats used are need to be mentioned.
- The conclusion is too long, please concise the statement.
Author Response
Dear Reviewer,
Please find enclosed the revised manuscript Membranes-1483104 and Response to Reviewer.
Sincerely yours,
Virjinia Doltchinkova

Round 2
Reviewer 1 Report
Title: “ Surface properties of synaptosomes in the presence of L-glutamic and kainic acids: in vitro alteration of the ATPase and acetylcholinesterase activities”
In this work the authors measured decrease in the bending modulus of model bimolecular membranes composed of monounsaturated lipid 1-palmitoyl-2-oleoyl-sn-glycero-3-phosphocholine provides evidence for softer membranes in the presence of L-glutamate. The authors claim that kainic acid does not affect the membrane mechanical stability even at tenfold higher concentrations. The authors presumed that both the L-glutamic and kainic acids reduce acetylcholinesterase activity and deviation from normal functions of neurotransmission
in synapses. Finally, the authors claim that the presented results regarding the modulation of the enzyme activity of synaptic membranes and surface properties of synaptosomes are expected to contribute in elucidation of the molecular mechanisms of neurotransmitters/agonists’ action on membranes.
General comment: The authors partially revised this work. As a consequence some issues should be still clarified within the current version of the manuscript.
Some detailed comments:
Lines:"
2.7. Thermal shape fluctuation analysis of GUVs
285 According to the theoretical model describing the mechanical properties of lipid bi-
286 layers [49, 50] the elastic energy per unit area, Fc, is given by the expression:
287
?
= ?
288 ?? + ? – ?)2/ 2 + ??? (4)
289
290 where kc represents the measured quantity, namely the bending modulus, c1 and c2
291 denote the principle modulus, c0 is the spontaneous curvature and kGstands for the Gauss-
292 ian curvature modulus of the bilayer. The contribution of the latter vanishes in the case of
293 GUVs’ closed membranes. As far as in the present study we consider symmetrical bilayers
294 with zero spontaneous curvature, ? = 0. Thermal shape fluctuation analysis (TSFA) of
giant lipid vesicles is applied to assess the bending modulus, kc, of lipid bilayers in the
296 presence of 10 mM of the neurotransmitters studied [51-53].
297 The visualization of the membrane fluctuations is carried out by a phase-contrast in-
298 verted microscope (Axiovert 100, Zeiss, Germany) equipped with a two-channel thermo-
299 control (±0.1°C) system operating both with the microscope plate and the oil-immersed
300 objective (100x, numerical aperture 1.25). The image-recording is performed by means of
301 a CCD camera (C3582, Hamamatsu Photonics, Japan) and a frame grabber (DT3155, Data
302 Translation, USA, 768x576 8-bit pixels, pixel size: 0.106 μm/pix) mounted in a Linux-op-
303 erating station for digitization and analysis [51-54]. The stroboscopic illumination of the
304 observed vesicles, synchronized with the camera, permitted the fast modes of fluctuations
305 to be recorded and processed [54]. All measurements of the bending elasticity are carried
306 out at (37±0.1) °C. Only visibly fluctuating quasispherical vesicles larger than 10 μm in
307 diameter are chosen for TSFA. Several hundred images are captured once per second and
308 subjected to TSFA for calculation of the membrane bending constant and tension of every
309 vesicle studied [51-53]. The reported values of the membrane bending modulus are ob-
310 tained over an ensemble of not less than five GUVs collected from at least three different
311 electroformation batches
*) This part seems to be crucial for the evaluation of the bending elasticity of the lipid membranes in the presence of L-glutamate and kainic acid.
This is a quite complex experimental procedure. Nevertheless, it seems that within the current version of the manuscript a sufficiently clear, detailed and quantitative description of the results related to the thermal shape fluctuation analysis (TSFA) lacks within the "Results" section.
As a consequence, the authors should provide a more detailed description, together with relevant images, calculations results and statistical treatment results, within the "Results" section. Similarly, they should discuss all this added material within the "Discussion" section.
All this material should be already available for authors (who used it to provide their main claims about the change of bending elasticity the lipid membranes in the presence of L-glutamate and kainic acid), thus they only have to add the previous material to better present and quantitatively explain their main claims to the interested readers (no novel experiments seem to be needed).
Lines:"3.4. Bending elasticity of model lipid membranes in the presence of L-glutamate and kainic acid
469 Control measurements of POPC membrane bending rigidity are performed both in
470 bidistilled water and in 50 mM Tris-HCl buffer. The obtained results indicate that the
471 bending modulus is not significantly altered in the buffer solution compared to its value
472 in water (Table 1 and Figure 6). The electroformation in 10 mmol/L of L-Glu yields small
473 vesicles and foam-like structures non-appropriate for TSFA. Hence, we prepare GUVs in
474 buffer containing ten-fold lower content of the neurotransmitter, namely 1 mM L-Glu. In
475 the latter milieu the electroformation protocol produces well fluctuating quasispherical
476 vesicles, allowing for shape fluctuation analysis and deduction of the membrane bending
477 modulus.
*) The authors should improve this section with the novel addiction of material (images and quantitative results) previously described (see previous comment).
Author Response
Dear Reviewer,
Please find enclosed our revised manuscript Membranes-1483104, final version of the manuscript Membranes-1483104 and revised Figures, Table 1 and Supplementary Material.
Sincerely yours,
Virjinia Doltchinkova

This manuscript is a resubmission of an earlier submission. The following is a list of the peer review reports and author responses from that submission.
Round 1
Reviewer 1 Report
This article entitled “Surface properties of synaptosomes in the presence of L-glutamic and kainic acids: in vitro alteration of the ATPase and acetylcholinesterase activities” by Doltchinkova describes several enzyme activities and membrane properties of synaptosomes incubated with different neurotransmitters/agonists. Unfortunately, the methods for synaptosome preparation are scarcely stated, and many of the results shown here indicate no systemic changes, which makes it difficult to evaluate their reproducibility. Overall, this study fails to provide sufficient information that benefits the readers.
Major points
The aim of the study is not clear. For example, in the abstract (L14), the authors stated “synaptosomes in brain in vivo”. Does this mean the authors believe there are synaptosomes in the brain? If so, it is a profound misunderstanding.
Related to the concern above, in the brain, dynamics of lipids in the neuronal membrane allows a rapid exchange in the composition of synaptic membranes in contrast to small particles like synaptosomes. It is not clear how this study is relevant to in vivo physiology.
According to the authors, synaptosomes are prepared from grey matter of rat brains. This is rather peculiar description. Assuming it is taken from cortex, which part of cortex, from what age etc. should be described. Also, how do you dissect out grey matter from white matter?
Dependent on which region of brain the synaptosomes are prepared, contents of enzymes studied here should vary a lot. Any conclusions drew here, such as “kainic acid show stronger effect acetylcholinesterase than glutamate” is not useful.
Minor point
Kainic acid is an agonist for a specific ionotropic glutamate receptor subtype. Although it could cause excitotoxicity (so as glutamate!), it is not a neurotoxin.
Author Response
Dear Reviewer,
Please find enclosed our response to your comments.

Reviewer 2 Report
This manuscript addresses an important issue about the modulation of the electrokinetics and enzyme activity of synaptosomal membranes upon the action of signaling molecules such as glutamate and KA. The results are clearly presented and the proposed mechanism is quite convincing.
I think some minor revision of manuscript is needed before it can be accepted for publication, which I believe will improve the readability of the paper. Additionally, the manuscript would benefit from language editing by either a native English speaker or a professional editor.
- There are many places where there are no spaces, so it needs to be corrected. (page7 line295 etc.)
- In Figure1, “a significant increase in enzyme activity is observed,” so significance probability and asterisks should be included in the Figure.
- In Figure 3-5, the explanation about the asterisks and number of experiments should be written in the legends.
- In Figure 6, is there any significance between control and L-Glu?
Author Response

(The authors gave the same response as above.)

Reviewer 3 Report
Title: “Surface properties of synaptosomes in the presence of L-glutamic and kainic acids: in vitro alteration of the ATPase and acetylcholinesterase activities “
In this study the authors investigated the electrostatic interaction of L-Glu and KA with the synaptosomal membranes, as well as the electrokinetic properties of synaptosomes. The authors aimed at determining the relationship between the surface electrical charge of the synaptosomal membrane and its stability during in vitro aging, together with the membrane rigidity.
General comment: It seems that a great amount of experimental work has been done in this research, and this is laudable. However, some part of this work are not clear because of the quality of the language, which does not allow the interested readers to follow the general meaning of the work. Indeed, some sections are not clear (e.g. the Abstract and part of the Introduction) thus interested readers can not fully appreciate the quality of the work. Finally, the value of this work is not clearly described and it should be better explained within the “Conclusion” section.
Some detailed comment:
2.7. Thermal shape fluctuation analysis of GUVs
Lines: “where kc represents the measured quantity, namely the bending modulus, c1 and c2 denote
the principle curvatures, c0 is the spontaneous curvature and kG stands for the Gaussian curvature of the bilayer”
*) The authors should add a scheme to better explain the meaning of C1,c2,c0 and Kg to the interested readers.
lines: “The reported data is
575 expected to shed light on the path towards further revealing the molecular mechanisms
576 of nerve cell activity. The latter is considered as a prerequisite for fostering the devel-
577 opment of novel therapeutic approaches in the treatment of neurodegenerative condi-
578 tions. “
*) The authors should better explain these lines and better underline the value of this work providing clear examples of application of the main results of this study.
*) Some strange words along the main text (e.g. “orphysical“ ,etc) should be corrected.
Author Response

(The authors gave the same response as above.)

Round 2
Reviewer 1 Report
Unfortunately, the authors did not seem to understand that the magnitude of variety of synapses in the brains. Overgeneralized data cannot be useful for anyone. The physiological differences should be tied to different regions of brain for synaptosome preparations so they can compare and discuss. The design of study has to be rebuilt.